# Why Do Inverse Eddy Surface Temperature Anomalies Emerge? The Case of the Mediterranean Sea

Evangelos Moschos *[ID], Alexandre Barboni and Alexandre Stegner [ID]

Laboratoire de Météorologie Dynamique, CNRS-IPSL, Ecole Polytechnique, 91128 Palaiseau, France
* Correspondence: evangelos.moschos@polytechnique.edu

**Abstract:** It is widely accepted that the signature of anticyclonic (cyclonic) eddies on the sea surface temperature corresponds to a warm (cold) core anomaly. Nevertheless, this statement has been put to question by recent regional studies showing the existence of inverse eddy SST anomalies: Cold Core anticyclones and, respectively, Warm Core cyclones. This study shows that the emergence of these inverse anomalies is a seasonal phenomenon that affects the life cycle of mesoscale eddies in the Mediterranean Sea. We use remote sensing observations and in situ data to analyse the eddy-induced SST anomaly over a 3-year period (2016–2018). We build an eddy core SST anomaly index to quantify the amount of Cold Core anticyclones and Warm Core cyclones all over the year and especially during the spring re-stratification period. We find that 70% of eddy anomalies are inverse in May and June both for cyclones and anticyclones. Regular temperature anomalies could reach 1.5 °C, while inverse ones are only present in the first 50 m of the oceanic layer and hardly exceed 1 °C. In order to understand the underlying dynamical processes, we construct a simple vertical column model to study the impact of the seasonal air–sea fluxes on the surface stratification inside and outside eddies. It is only by taking into account a differential diapycnal eddy mixing—increased in anticyclones and reduced in cyclones—that we reproduce correctly, in agreement with the observations, the surface temperature inversion in the eddy core. This simplified model suggests that vertical mixing modulation by mesoscale eddies might be the key mechanism that leads to the eddy–SSTA seasonal inversion in the ocean.

**Keywords:** ocean mesoscale eddies; sea surface temperature; vertical mixing

## 1. Introduction

Mesoscale eddies are coherent structures with typical radii of the order of tens up to a hundred kilometres and timescales on the order of a month. These eddies can be sometimes long-lived, surviving several months or even years. Significant advances in the resolution of both satellite altimetry measurements [1] and high-resolution oceanic numerical models [2] have revealed the predominance of these mesoscale eddies in the global oceanic circulation. They are able to trap and transport heat, salt, pollutants and various biogeochemical components from their regions of formation to remote areas [3,4]. Eddies are formed through shear and meander instability of boundary currents [5], baroclinic instability [6], the effect of wind on the sea surface [7] and other mechanisms. Their dynamics can impact significantly the biological productivity at the ocean surface [8–10], modify the depth of the mixed layer [11], influence clouds and rainfall within their vicinity [12], amplify locally the vertical motions [13], attract pelagic species [14–16] or concentrate and transport microplastics [17]. Long-lived mesoscale eddies are ubiquitous in the global ocean and play a major role in its circulation differentiating from mean patterns. In the Mediterranean Sea, the domain of this study, mesoscale eddies have been identified, tracked and analysed, both on their surface and subsurface structure in many studies [7,18–24].

The use of infrared images, which measure the Sea Surface Temperature (SST), has allowed the detection of many oceanic eddies and a better understanding of regional circulations [18,25,26]. These detections were performed visually by expert oceanographers.

However, due to the scarcity of in situ observations, it was not until the intensive development of satellite altimetry and the development of automatic vortex detection algorithms on Sea Surface Height (SSH) [1] that a statistical link between Eddy-induced Sea Surface Temperature Anomalies (eddy-SSTA) and SSH anomalies was established.

Several studies working on SST composites of eddies detected on the SSH associate Warm Core eddies (positive eddy-SSTA) with anticyclones and Cold Core eddies (negative eddy-SSTA) with cyclones [12,27,28]. However, through the individual analysis of eddies, various regional studies have shown the existence of inverse temperature anomalies, i.e., anticyclones (respectively cyclones) with a Cold (Warm) core anomaly. In a study of Mediterranean circulation using SST data, Ref. [18] performed observations of some Cold Core anticyclones on the summer period in the Eastern Mediterranean sea. In the Tasman Sea, Ref. [29] observed the existence of an important fraction (70%) of inverse anomalies. In the southwestern Atlantic Ocean, [30] found cyclones with a warm eddy-SSTA, which are explained through their (warm) region of formation. In the Arabian Sea, Ref. [31] showed the existence of inverse anomalies while searching for a link between the SST and Mixed Layer Depth (MLD) anomaly. In the North Pacific Ocean, Ref. [32] performed similar observations of inverse anomalies and also showed a seasonal variability in the regional eddy temperature anomaly distribution, noting that these inverse anomalies appear for shorter times than the regular ones. In the same fashion, Ref. [33] analysed the inverse eddy-SSTA in the South China Sea and noted a slight dependence on both seasonal effects and eddy amplitude. The last two studies both link inverse anomalies with the summer re-stratification at the ocean surface. Furthermore, Ref. [34] build an index based on the SST anomaly of an eddy to distinguish between surface and subsurface structure.

The presence of Cold Core anticyclonic and Warm Core cyclonic eddies on a global scale has also been documented by two recent studies. Through a Deep Learning eddy identification method, Ref. [33] detected and classified eddies and their surface temperature anomaly. An important fraction of inverse anomalies is revealed around the globe, reaching up to 40%. The authors also showcase the seasonal variation of this fraction as well as an inter-annual trend of diminishing inverse anomalies. In the same manner, [35] showcases that inverse anomaly eddies have lower absolute eddy-SSTA values than their regular counterparts. Exhibiting strong seasonal variation, inverse anomalies cover according to this paper 15% of anticyclones (10% cyclones) in the summer period. Finally, the authors show a correlation of this seasonal variation of eddy SST anomalies with the mixed layer modulation, along with the inversion of wind-stress and heat-flux patterns over these eddies. It should be noted that the percentages of inverse anomalies differs significantly between the aforementioned studies (regional and global) based on the method used to quantify them.

However, correlation does not imply causation, and even if some of the aforementioned articles create a strong observational link, regionally or globally, between the mixed layer modulation and the inversion of eddy-SSTA, none of them demonstrates an underlying mechanism for this phenomenon.

The goal of this work is to perform a comprehensive study on the formation of inverse sea surface temperature anomaly of mesoscale eddies and propose an underlying physical mechanism. As a case study, observations in the Mediterranean Sea are examined, although our results can be expanded to other regions of the globe. Here, we attempt to answer four questions:

- *How does the eddy-SSTA distribution vary seasonally?* We first define an eddy core surface temperature anomaly index to quantify the intensity of the eddy-SSTA for a large number of anticyclonic and cyclonic eddies in the Mediterranean Sea. This index allows us to perform a statistical analysis of the seasonal variations of the temperature anomaly inside coherent eddies and study its correlation with the evolution of the MLD.
- *How does the SST signature and anomaly of an individual mesoscale structure evolve?* We investigate a few long-lived eddies to follow the temporal evolution of their SST

anomaly with respect to their dynamical parameters and the seasonal stratification of the ocean surface.

- *Is the surface temperature anomaly linked with the subsurface structure ?* We quantify more precisely the evolution of the surface stratification inside and outside these selected eddies using ARGO profiles to estimate the eddy vertical temperature structure and compare it with the surface temperature anomaly.
- *Why do inverse SST anomalies emerge?* We propose a mechanism based on differential vertical mixing between the eddy core and its periphery under atmospheric fluxes, which is illustrated with idealised single-column numerical simulations. The relevance of this physical model to explain the inverse emergence of inverse eddy-SSTA and its agreement with the remote-sensing and in situ observations are discussed in the conclusion.

## 2. Materials and Methods

### 2.1. Satellite and In Situ Data

This study focuses on the mesoscale oceanic eddies of the Mediterranean Sea, during the three-year period 2016–2018. To perform our analysis, we combine satellite and in situ data to characterise both the ocean surface and the subsurface stratification. The infrared satellite imagery provides the SST maps which are the core data of this study. We use the DYNED-Atlas database to obtain the dynamical parameters and the contours of mesoscale eddies detected on standard satellite altimetry products. The three-dimensional structures of the studied eddies as well as the surface stratification and the MLD were derived from the in situ Argo floats measurements.

#### 2.1.1. Satellite Data

Daily, high-resolution (1/120°) super-collated SST maps of the Mediterranean Sea are received from the Copernicus—Marine Environment Monitoring Service (CMEMS), Ultra High Resolution L3S SST Dataset (https://doi.org/10.48670/moi-00171, accessed on 28 June 2022), produced by the CNR—Italy and distributed by CMEMS. The process of supercollation uses SST measurements derived from multiple sensors, representative of nighttime SST values [36]. Sea Surface Height (SSH) and geostrophic velocity fields, used for the detections of eddies in this study, are L4 AVISO/CMEMS altimetric products at 1/8° resolution retrieved from the CMEMS L4 Sea Level dataset. (https://doi.org/10.48670/moi-00141, accessed on 28 June 2022)

#### 2.1.2. Eddy Contours, Centers and Tracks

The dynamical evolution of eddies and their individual tracks are retrieved from the DYNED-Atlas database for the three year period 2016–2018. The DYNED-Atlas (https://www1.lmd.polytechnique.fr/dyned/, accessed on 28 June 2022) project containing eddy tracks and their physical properties is publicly accessible. The tracking of these eddies is performed by the AMEDA eddy detection algorithm [37] applied on daily geostrophic velocity fields derived from the AVISO/CMEMS SSH maps. A cyclostrophic correction is applied on these geostrophic velocities to accurately quantify eddy dynamical properties [38]. Unlike standard eddy detection and tracking algorithms, the main advantage of the AMEDA algorithm is that it detects the merging and the splitting events and allows thus for a dynamical tracking of eddies [37].

The identification of potential eddy centers by AMEDA is performed by computing the Local Normalised Angular Momentum (LNAM) [21] of the geostrophic velocity field. Only eddy centers with at least one closed contour of the stream function of the velocity field are retained. A radial profile of the velocity for each detected eddy center is calculated by computing the average velocity and radius at each closed streamline around it:

$$\langle V \rangle = \frac{1}{L_p} \oint \vec{V} \, d\vec{l} \tag{1}$$

where $V$ is the local geostrophic velocity field and $L_p$ is the streamline perimeter. The maximum velocity, obtained through Equation (1), will be hereby noted as $V_{max}$ and the radius corresponding to the characteristic contour. The radius $R_{max}$ of the characteristic contour is obtained by considering a circular contour of an equivalent area $A$:

$$\langle R \rangle = \sqrt{\frac{A}{\pi}} \qquad (2)$$

The eddy centers and their characteristic radius $R_{max}$ are important parameters used to retrieve SST patches for each eddy detection.

### 2.1.3. Argo Floats

Hydrographic profiles of Argo floats are received through the CORIOLIS program database (http://www.coriolis.eu.org/, accessed on 28 June 2022). Potential temperature and salinity profiles are received from Argo floats through which the potential density profiles are derived. A co-localisation is performed between the position of Argo floats and the detected eddies of the DYNED-Atlas database. Argo profiles are marked as inside an eddy if their distance $r$ from any eddy center is $r < R_{max}$ and outside an eddy if the above condition is false for every eddy detection of the same day.

To calculate the MLD of each Argo observation, we use its potential density profile and search for the maximum depth at which a threshold of $\delta\rho = 0.03 \text{ kg/m}^3$ is reached [39].

### 2.2. A Method to Quantify Eddy-Induced SST Anomalies

Mesoscale eddies often have a visible signature on SST images, with a temperature difference between the waters in the eddy core and the waters in its periphery. This difference is defined as the eddy-induced surface temperature anomaly (eddy-SSTA) of an eddy, and it can be quantified through our proposed methods.

A procedure to retrieve a large dataset of SST image patches containing eddy signatures is proposed, following [40,41]. The Eddy-Core Surface Temperature Anomaly Index ($\delta T$), a simple and heuristic method for quantifying the eddy-induced temperature anomaly represented in each image, is then defined. The proposed methodology, applied here to observations in the Mediterranean Sea, is generic enough to provide results in every oceanic domain.

### 2.2.1. Eddy SST Patches Dataset

A thorough statistical analysis of eddy-induced SST anomalies requires a large dataset of SST image patches in the Mediterranean Sea. The characteristic contours (highest mean velocity) of the mesoscale eddies contained in the DYNED-Atlas are used to crop patches from SST maps, which are centred on the detected eddies. These contours can either represent Anticyclonic Eddies (AE) or Cyclonic Eddies (CE) rotating clockwise and anti-clockwise, respectively, in the northern hemisphere. For each eddy, a square patch of size ($5R_{max} \times 5R_{max}$) is cropped and centred on the contour barycenter. Cloud coverage creates missing values on satellite SST images and can corrupt the signature of the cropped image patches. Thus, only patches with less than 50% of cloudy pixels are retained.

The eddy SST signatures can be distinguished either as Warm Core or Cold Core anomalies, as discussed earlier. Four such cases are depicted in Figure 1 in which both positive and negative SSH anomalies can correspond to a Warm or a Cold SST anomaly. The characteristic contours of each eddy (in black) are superimposed on the Absolute Dynamic Topography (ADT), derived from the SSH, and the corresponding SST patch.

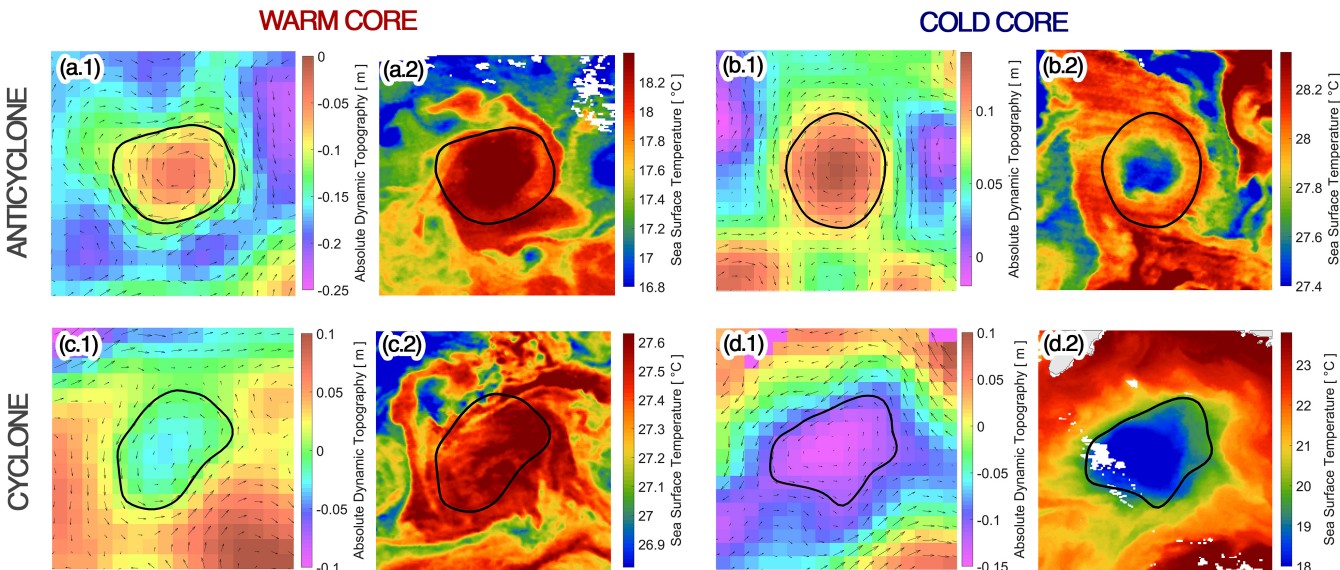

**Figure 1.** Samples of eddy-induced SST anomalies for a (**a**) Warm Core Anticyclone, (**b**) Cold Core Anticyclone, (**c**) Warm Core Cyclone, (**d**) Cold Core Cyclone. On the leftmost panels numbered with (**1**), the velocity vectors and the characteristic contour, computed by the AMEDA algorithm (black line), are superimposed on the Absolute Dynamic Topography. On the rightmost panels numbered with (**2**), the characteristic contour (black line) is superimposed on the patches of Sea Surface Temperature field centred on the detected eddy. Image patches are of side $5R_{max}$, which are chosen in order to include the temperature of waters.

2.2.2. The Eddy-Core Surface Temperature Anomaly Index [$\delta T$]

The Eddy-Core Surface Temperature Anomaly Index (hereby $\delta T$) is a simple and heuristic metric of the temperature difference between the core (centre) of the eddy and its periphery. We define the core of the eddy as the region enclosed by the maximum velocity contour [37]. The value of $\delta T$ is calculated as the difference between the mean of the temperature values inside a *core-mean frame* and a *periphery-mean frame* in a given patch, with units in °C. These two square frames, which share a common centre, have sides of $R_{max}$ and $5R_{max}$, respectively. For the calculation of the mean value in the periphery-mean frame, the values contained in the core-mean frame are ignored. A positive $\delta T$ value denotes a larger core-mean temperature than its periphery-mean temperature and thus a Warm Core Eddy, while a negative $\delta T$ denotes, respectively, a Cold Core Eddy. The calculation of the $\delta T$ variable by use of the core-mean and periphery-mean frames is shown in Figure 2. Examples (a) and (b) show the two centred anomalies, which are shown also in Figure 1a.2,b.2. The $\delta T$ values are 0.75 °C for the Warm Core example (a) and −0.27 °C for the Cold Core example (b).

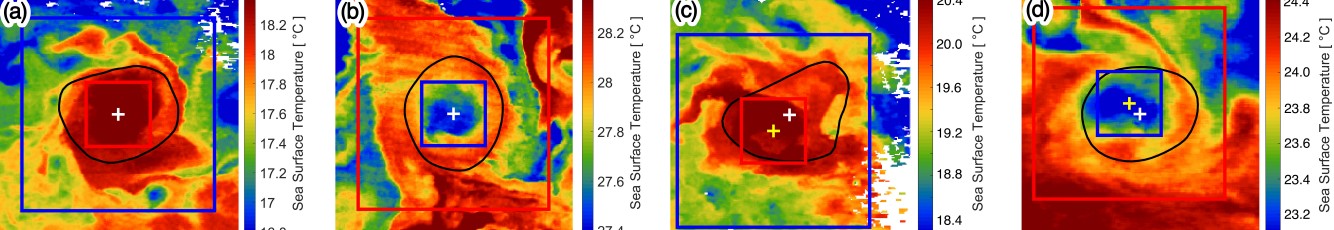

**Figure 2.** Examples of the eddy-core surface temperature anomaly computation and offset method. Snapshots represent Sea Surface Temperature in degrees. Black lines are superimposed altimetric detection contours. Small squares represent the core-mean and large ones represent the periphery-mean frames. Core-periphery values are coloured red-blue or blue-red based on the sign of $\delta T$. Examples (**a**–**d**) illustrate the correction by offset: A white cross marks the centre of the image. A yellow cross marks the center of the core-mean and periphery-mean frames, if it differs from the centre of the image. Examples (**a**,**b**) are centred while (**c**,**d**) are offset.

However, the barycenter of the velocity contour can differ from the centre of the eddy SST anomaly core, due to bias or errors of altimetric maps [42,43]. Therefore, an offset of both frames is considered in order to locate the exact position of the maximum eddy-SSTA and correct the index value.

This correction is computed as follows: First, the value of $\delta T$ is calculated through squares centred on the image, as described above. The sign of the eddy-induced SST anomaly is thus defined. Then, if $\delta T$ is positive (negative), the warmest (coldest) core-mean value is searched for by offsetting the core-mean frame in all directions with a stride of $\frac{1}{9}R$ and a maximum offset of $\frac{2}{3}R$. Finally, the periphery-mean frame is centred along the shifted core-mean frame, and the corrected $\delta T$ value is computed. In the rest of this manuscript, $\delta T$ represents the final values calculated by applying the offset correction.

Examples of off-centred eddy detections are shown in Figure 2c,d. The core and periphery have been shifted in order to maximise the eddy-core surface temperature anomaly index. The geometric centre of the image is shown with a white cross, while the shifted centre of the core-mean frame is shown with a yellow cross. The $\delta T$ values are 0.68 °C corrected to 0.86 °C by offseting for the Warm Core example (c) and $-0.46$ °C corrected to $-0.55$ °C by offseting for the Cold Core example (d).

Nevertheless, even with this correction, a significant amount of noisy and/or corrupted SST signatures remain. This could be due to the combination of erroneous eddy detections on gridded AVISO/CMEMS altimetry products, large-scale air–sea interactions that mask the mesoscale eddy signature or the presence of clouds [41].

In order to exclude these images with unclear SST signatures, two thresholds are considered. The Cloud Coverage threshold, described above, is used to retain only images that have a Cloud Coverage Percentage (CCP) lower than 50%. The CCP is defined as the percentage of pixels covered by clouds on a given area. This criterion is applied twice: on the whole image patch ($CCP_{patch}$) as well as the core-mean frame ($CCP_{frame}$). The threshold is chosen so that the eddy SST signature is not corrupted, which could produce errors in the calculated $\delta T$ value [41].

An illustration of the application of the Cloud Coverage threshold is provided in Figure 3a–d, where snapshots of the SST signature of the same eddy (Ierapetra) are provided at different days of December 2016 along with core-mean and periphery-mean frames. Example (a) on 19/12 shows a patch with an overly clear eddy signature ($CCP_{patch} = 8\%$, $CCP_{frame} = 0\%$) retained in the dataset. Example (b) on 20/12 shows a patch with an eddy signature covered by clouds ($CCP_{patch} = 40\%$, $CCP_{frame} = 48\%$), which however does not surpass the 50% threshold and is retained in the dataset. Examples (c) on 29/12 ($CCP_{patch} = 48\%$, $CCP_{frame} = 90\%$) and (d) on 30/12 ($CCP_{patch} = 72\%$, $CCP_{frame} = 76\%$) show patches exceeding the Cloud Coverage threshold and therefore filtered from the dataset.

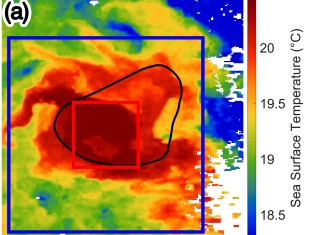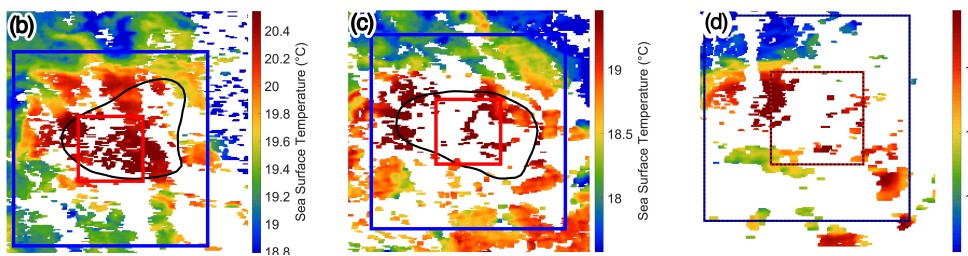

**Figure 3.** Examples with different Cloud Coverage. Snapshots represent Sea Surface Temperature in degrees. Contours and squares are the same as Figure 2. Snapshots of the same eddy (Ierapetra) on different days of December 2016 (**a**) 19/12, (**b**) 20/12, (**c**) 29/12, and (**d**) 30/12. Cloud coverage percentage is increasingly high. Examples (**a**,**b**) are retained, while (**c**,**d**) are not retained

Finally, a filter on weak $\delta T$ values is also applied. We have noticed by visual inspection that unclear SST signatures often induce a weak value of the $\delta T$. Hence, to filter out these noisy images, we retain only SST patches if $|\delta T| > 0.1$.

## 3. Results

### 3.1. Seasonal Variability of the Eddy-Induced Temperature Anomaly

The seasonal variability of the eddy-induced temperature anomaly, and more generally the signature of eddies on the SST, is analysed in this section through two different perspectives: first, a statistical analysis is carried out on the $\delta T$ values calculated on every retained eddy detection. Furthermore, the change of the surface temperature anomaly is examined over the lifetime of several long-lived eddies in the Mediterranean Sea while particularly focusing on the subsurface structure of one of them.

#### 3.1.1. Statistical Analysis

Composite averages are often employed in the bibliography to represent the SST anomaly of mesoscale eddies [12,27,28]. This averaging leads frequently to the association of a Warm Core anomaly to anticyclonic eddies and a Cold Core anomaly to cyclonic eddies. To examine these average temperature anomalies, we calculate here the composites of all eddy SST patches retained after first performing a normalisation per patch. To receive the Normalised SST Anomaly, we subtract from each pixel the mean value of all the SST values of the entire patch and divide the result by the standard deviation of all the SST values of the entire patch. In Figure 4, composites of Anticyclonic and Cyclonic Normalised SST Anomaly are plotted for all eddies and those observed on the Winter (December–January–February) and Summer (May–June–July) period. These two oceanic seasons are chosen on the three-month period when the mean value of the MLD, computed outside the detected eddies, reaches its largest or smallest value .

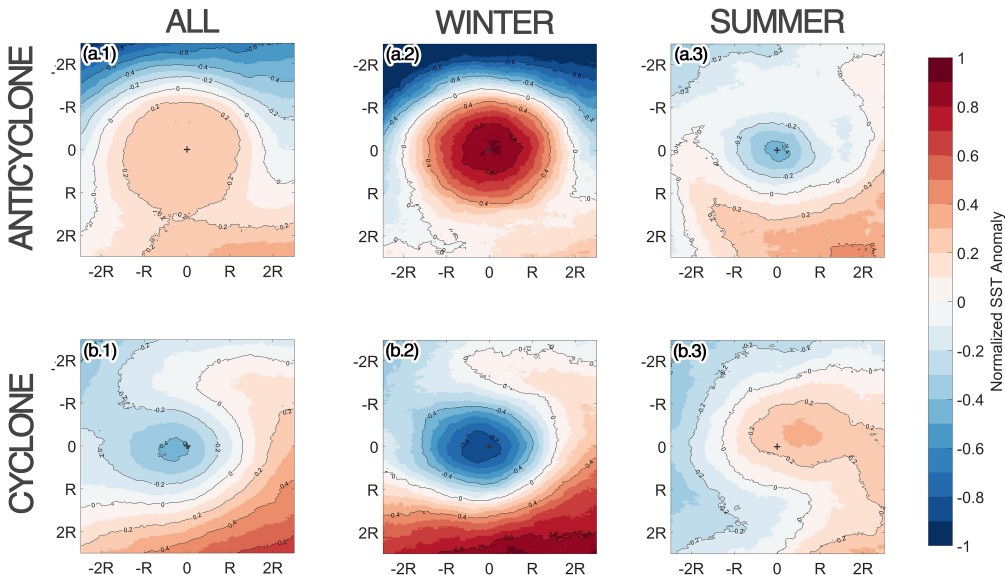

**Figure 4.** Composite averages of normalised SST anomaly for (**a**) Anticyclonic and (**b**) Cyclonic eddies for (**1**) all, (**2**) winter (DJF), and (**3**) summer (MJJ) observations. Each value in an eddy SST patch is normalised by subtracting the mean value and dividing by the standard deviation of all values. Composites are retained by averaging between patches.

From the composites of Figure 4, it can be seen that the average SST anomaly of all anticyclonic (cyclonic) observations indeed corresponds to a Warm (Cold) Core structure, or else the regular eddy anomaly. Nevertheless, a strong seasonal variation of this average anomaly is revealed by plotting the winter and summer composites. In winter, the regular anomaly is even more pronounced with double to triple normalised anomaly values. However, summer composites show an inverse average anomaly, i.e., Cold Core Anticyclones and Warm Core Cyclones on average, while also having weaker normalised anomaly values. The latter is coherent with the findings of other studies, showing that the SST signatures on the vicinity of eddies on summer tend to be more spatially uniform [35,44].

While composites suffice to portray the seasonal inversion of eddy temperature anomalies, averaging out patch values does not retain the variance in eddy anomalies on the SST. To quantify the latter, we perform a statistical analysis of the $\delta T$ index values computed for all the patches retained.

The histograms of the $\delta T$ index are shown in Figure 5, separately for anticyclonic and cyclonic eddies, in winter (DJF) and summer (MJJ). On the histograms, red bins represent Warm Core observations, while blue bins represent Cold Core observations. Grey bins represent observations where $|\delta T| < 0.1$. These bins correspond to outlier values linked with the noise on the SST data as well as errors on the sensors observation and our method. A threshold of $|\delta T| > 0.1$ is fixed to filter out these observations in the analysis/figures that follow.

If we consider a year-long statistical distribution, AE are predominantly Warm Core and CE are predominantly Cold Core; in other words, AE and CE, exhibit on average an *regular* anomaly on the SST. However, the distribution of $\delta T$ values in the histograms of Figure 5 suggest that the eddy-core temperature anomaly exhibits strong seasonal variation, altering between Warm Core and Cold Core anomaly regimes. Specifically, during winter, the regular anomalies are preponderant: 93% of AE observations correspond to Warm Core eddy, while 92% of CE observations are Cold Core. However, during summer, Cold Core AE and Warm Core CE observations become dominant with, respectively, 69% and 66% of the observations. It is due to this seasonal inversion of the regular anomaly that we name the Cold Core AE and Warm Core CE as *inverse* SST anomaly.

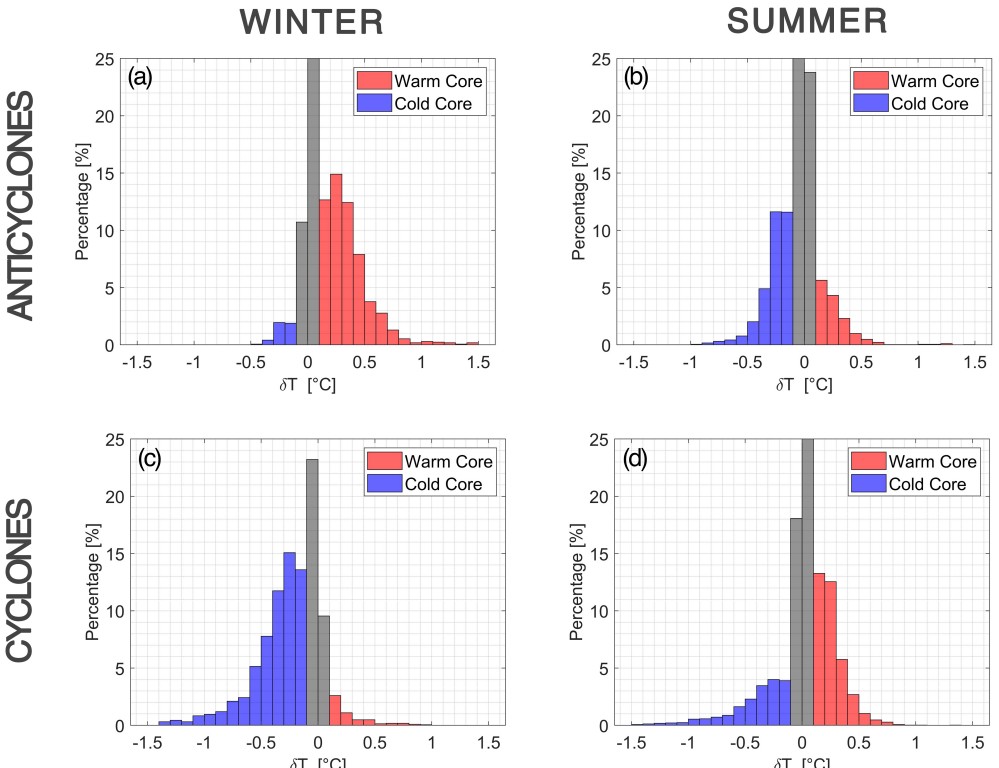

**Figure 5.** Seasonal histograms of $\delta T$ values. (**a**) AE in the winter mixing period (DJF), (**b**) AE in the summer restratification period (MJJ), (**c**) CE in the winter mixing period (DJF) and (**d**) CE in the summer restratification period (MJJ). Red bins represent positive $\delta T$ Warm Core observations, while blue bins represent negative $\delta T$ Cold Core observations. Grey bins represent observations where $|\delta T| < 0.1$.

The seasonal cycle of the eddy-SSTA of both AE and CE, in the Mediterranean Sea, coincides with the seasonal variation of the MLD. This is portrayed in Figure 6, where the monthly variation of the percentage of inverse eddy core anomalies is plotted along with the monthly variation of the MLD. The later is calculated as the mean of all Argo profiles that are located outside eddies. The winter mixing period (DJF), when the mean MLD is at its largest extent, coincides with the period when eddy anomaly are dominantly regular, with only 5–15% of inverse anomalies (i.e., 95–85% regular anomalies) for both AE and CE. Conversely, the end of the spring re-stratification period (MJJ) when the mean MLD is at its shallowest coincides with the period where most eddies have an inverse anomaly, reaching a peak of 70% of Cold Core AE and Warm Core CE observations for the months of May and June.

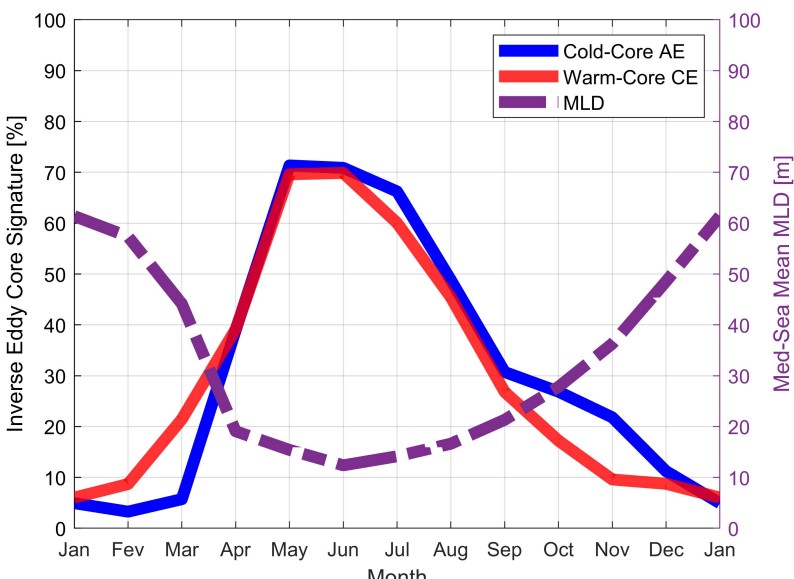

**Figure 6.** Seasonal variation of the mean MLD with inverse eddy anomaly percentage. For each month, the mean percentage of Cold Core AE observations is plotted with a blue line, the mean percentage of Warm Core CE observations is plotted with a red line, and the mean MLD of all Argo profiles located outside eddies with a dashed purple line.

The seasonal variation of the core anomalies and their summer inversion, seen on Figures 4–6, is also analysed spatially. In Figure 7, we plot the $\delta T$ values of all anticyclonic/cyclonic eddy detections in the Mediterranean Sea for one winter (DJF) and one summer (MJJ) season. The predominance of regular anomalies (Warm Core AE, Cold Core CE) in winter (panels a and c) and inverse anomalies (Cold Core AE, Warm Core CE) in summer (panels b and d) is spatially homogeneous over the Mediterranean Sea. Regular temperature anomalies reach higher absolute values, as can be seen by the intense red dots (i.e., Warm Core anticyclones on panels a and b) and blue dots (Cold Core cyclones on panels c and d). The inverse anomalies have lower absolute values, as portrayed in the histograms of Figure 5. Finally, Figure 7 also portrays a spatial homogeneity, with the emergence of inverse eddy anomalies on summer happening all over the Mediterranean Sea.

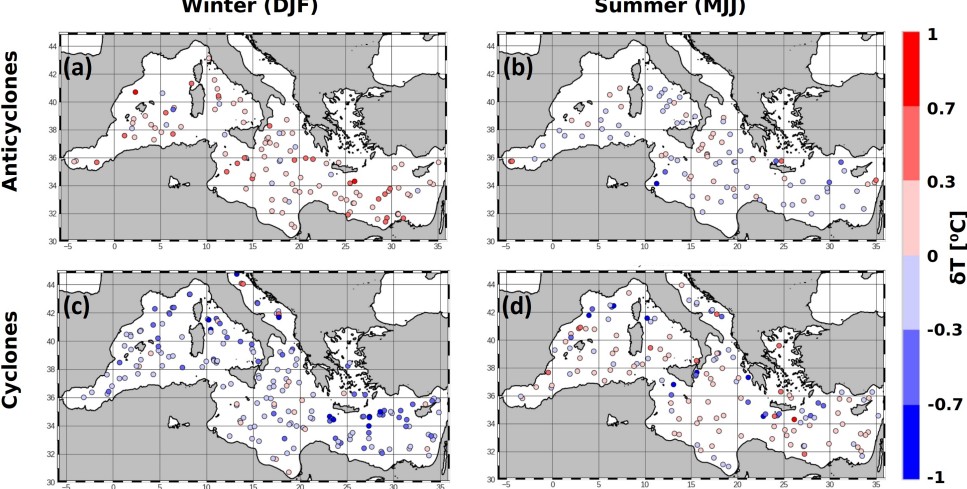

**Figure 7.** Spatial variation of $\delta T$ values in the Mediterranean Sea for (**a**) Anticyclones in winter, (**b**) Anticyclones in summer, (**c**) Cyclones in winter, and (**d**) Cyclones in summer. Red (Blue) dots correspond to warm (cold) anomalies. The colour intensity depicts the absolute value of the anomaly.

### 3.1.2. Individual Eddy Analysis

To better investigate how the seasonal evolution of the surface stratification inside mesoscale eddies impacts their surface temperature signature, we track four long-lived eddies and follow the temporal evolution of their dynamical parameters, the surrounding MLD and their surface SST anomaly. One of them, an Ierapetra Anticyclone, formed south of the island of Crete, was sampled for over a year by Argo floats trapped inside its core. Three more eddies are considered: a Cyprus anticyclone located among and around the Eratosthenes seamount, an Algeria Anticyclone drifting along the Algerian coast and an elongated cyclone located in the Ligurian sea. The timeline of the Ierapetra eddy is shown in Figure 8, while those of the Algeria, Cyprus and Liguria eddies are shown, respectively, in Figures A1–A3 of the Appendix A.

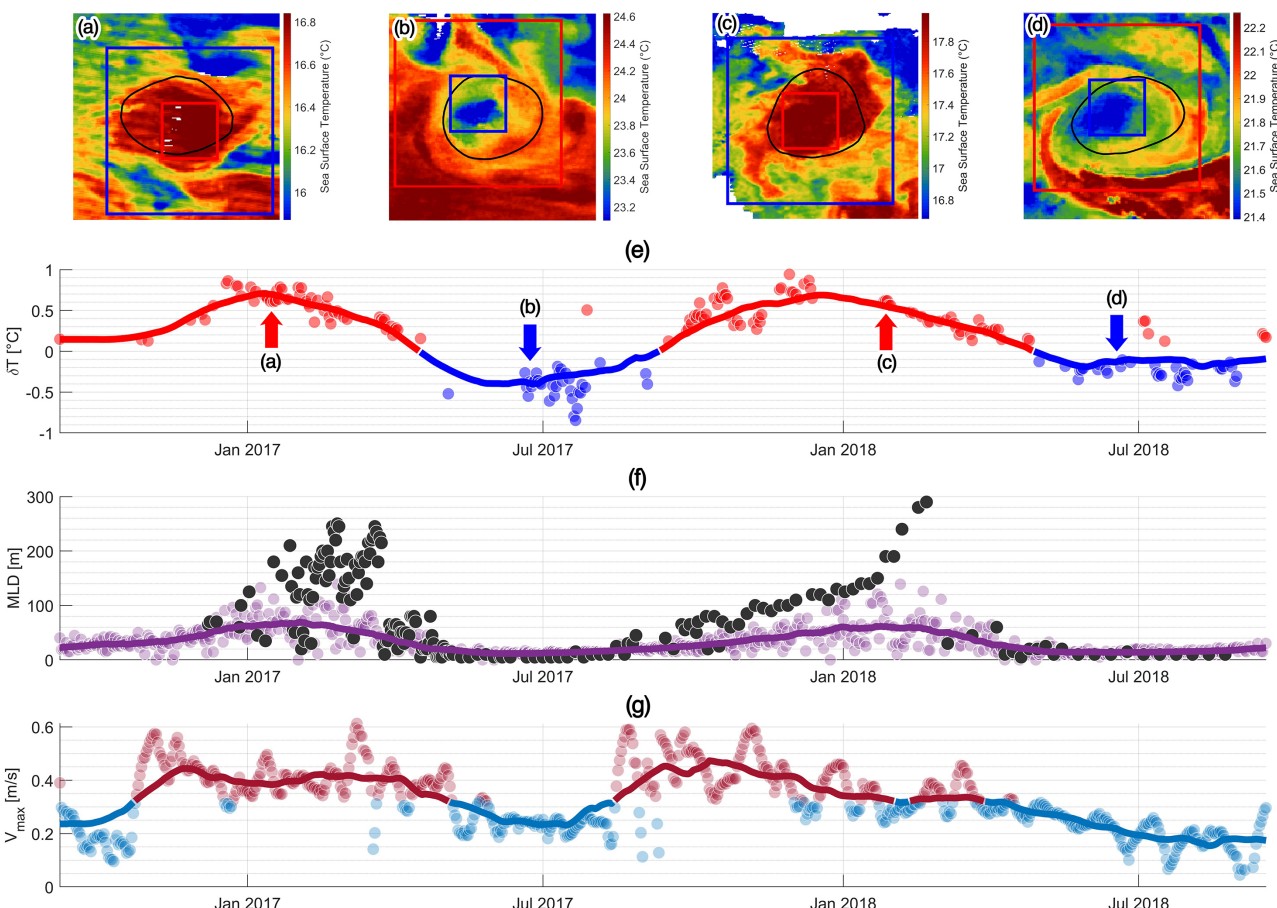

**Figure 8.** Timeline of the Ierapetra anticyclone with DYNED ID #11099. Four characteristic SST patches are shown in panels (**a**–**d**) which correspond to different regimes of the evolution of the eddy SST anomaly. In panel (**e**), the $\delta T$ index values are plotted for every retained observation with red (blue) dots for positive (negative) values. The Monthly Mean Average of these values is plotted with a red (blue) line showing the regime change between a Warm Core and a Cold Core eddy. On panel (**f**), the depth of the mixed layer (MLD) is plotted with pink dots for values outside the eddy and black dots inside the Ierapetra eddy. A Monthly Mean Average of the MLD evolution outside the eddy is plotted with a pink line. On panel (**g**), the surface maximal velocities ($V_{max}$) for each timestep in the DYNED-Atlas eddy track are plotted with dots and their Monthly Mean Average is plotted with a line. Velocities are plotted with magenta (cyan) when they are higher (lower) than the mean velocity in the eddy lifetime.

For each of these four eddies, we create an *Eddy Timeline* that contains the $\delta T$ index, the evolution of the MLD in the eddy area, the eddy intensity and a few characteristic snapshots of the eddy SST signature. Moreover, to highlight the seasonal variations, a Monthly Mean Average is calculated for each parameter. The daily values of the Core Temperature Anomaly Index ($\delta T$) are plotted when the cloud coverage is not too high and the temperature anomaly is not too small (i.e., $|\delta T| > 0.1$). The calculated ($\delta T$) (dots) as well as the corresponding Monthly Mean Average (line) are coloured in red (blue), when their value is positive (negative), denoting a warm (cold) core regime.

To construct the MLD time series (pink dots), we use all the Argo profiles that were measured outside eddy contours in a surrounding area, which are defined as a rectangular box of one degree of latitude and longitude and centred on the eddy. More than one MLD observation might exist for a certain day, causing a spread of values, especially for the winter mixing period. When in situ measurements are available inside the eddy contour, the MLD is plotted with black dots. The variations of the eddy intensity, quantified here by $V_{max}$, are plotted during the same period. In order to highlight the seasonal variations, we use distinct colours for the velocity above (magenta) and below (cyan) the mean velocity value averaged over the whole period.

Our analysis is focused on the evolution of an Ierapetra Anticyclone from September 2016 to September 2018. According to Figure 8, this anticyclone changes regimes twice from a regular to an inverse anomaly. The inverse Sea Surface Temperature anomaly begins in spring, when the re-stratification sets in, and continues until fall.

As can be seen in panel (f) of Figure 8, in winter months, while the eddy is in a Warm Core regime in panel (e), the MLD is two or three times deeper inside the Ierapetra anticyclone than in its surroundings, reaching 300 m of depth while being shallower than 120 m in its surroundings. The Warm Core surface anomaly of the eddy (panels (a) and (c)) can be linked therefore with its subsurface heat content. On the other hand, during the spring re-stratification period and a Cold Core regime, the MLD is rather shallow, not exceeding a few tens of meters both inside and outside the anticyclone. The Cold Core surface anomaly (panels (b) and (d)) is disconnected from the warm subsurface heat content. It should also be noted that it is during the winter months, when the MLD is deeper in the eddy core, that the anticyclone passes an intensification phase shown in the evolution of the surface velocity $V_{max}$.

Similar regime transitions from a regular to an inverse sea surface temperature anomaly are visualised in Appendix A of this article for two other anticyclones in Figures A1 and A2 as well as a cyclone in Figure A3. For all these eddies, the inverse Sea Surface Temperature anomaly is directly correlated to a small MLD in the area surrounding the eddy. This indicates a strong surface stratification on the same period, leading to a disconnection of the inverse surface anomaly with the subsurface heat content.

To investigate if the change in the surface core temperature anomaly is linked with the subsurface anomaly of the Ierapetra anticyclone, two profiles from an ARGO float released inside the core of the eddy are examined. From a series of observations, the profiles are chosen on two dates where the SST signature of the eddy is not corrupted by clouds and the in situ profile samples well the eddy core. In winter, a regular Warm Core observation on 26 February 2017 can be seen in panel (a.2) of Figure 9, corresponding to panel (a) of Figure 8. In summer, on 26 June 2017, we retain an inverse Cold Core observation of the same eddy that can be seen in panel (b.2) of Figure 9, corresponding to panel (b) of Figure 8. On these panels, a white star corresponds to the location of the eddy-sampling ARGO float. On panels (a.1) and (b.1) of Figure 8, the location of the eddy-sampling floats are shown with a star in a wider map, where we also plot the region (dashed rectangle) where we search for background sampling ARGO floats. We consider a $\pm 1$ week period from the observation date and search for ARGO profiles in that region that fall outside of eddies. The temperature measurements of these background outside-eddy profiles is plotted with dashed grey lines on panels (c) and (d) of Figure 8 while their mean is plotted with a thick black line. The eddy-sampling profile is plotted on panels (c) and (d) of Figure 8,

corresponding to the winter and summer periods, respectively. When the eddy-sampling profile is warmer (cooler) than the mean outside-eddy profile, it is plotted with a thick red (blue) line.

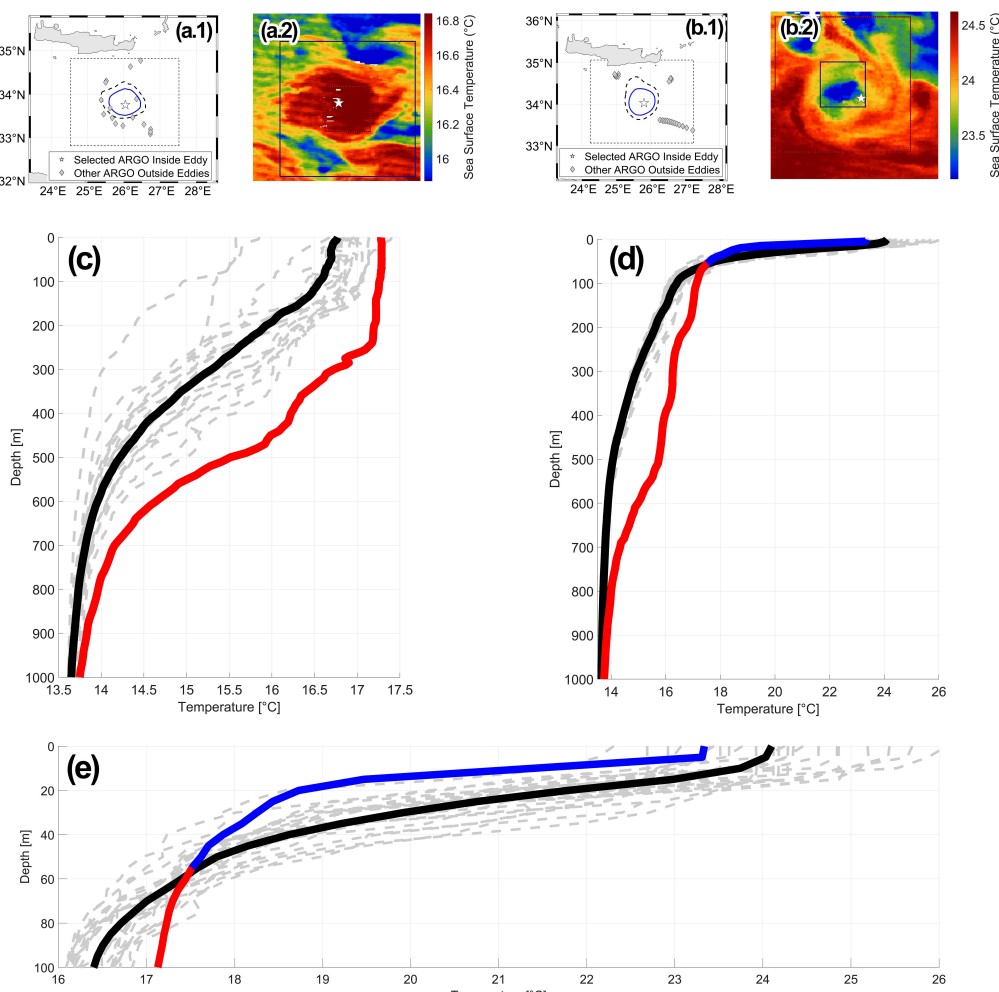

**Figure 9.** Seasonal evolution of the vertical structure of the Ierapetra eddy. Panels (**a**,**c**) correspond to a Warm Core SST observation of the eddy on 26/02/2017. Panels (**b**–**e**) correspond to a Cold Core SST observation on 26/07/2017. Maps (**a.1**,**a.2**) show the maximum velocity contour and outermost contour of the eddy, the eddy-sampling ARGO profile with a star and the outside-eddy profiles with diamonds, which are retained in a region outlined by the dashed rectangle. Patches (**b.1**,**b.2**) show the SST anomaly of the eddy along with the location of the eddy-sampling profile, which is plotted with a star. Vertical plots (**c**–**e**) show the outside-eddy profiles plotted with dashed gray lines and their mean outside-eddy profile with a thick black line. The eddy-sampling profile is plotted with a thick red (blue) line when it is warmer (colder) than the mean outside-eddy profile. Profile (**c**) shows the winter regular surface anomaly, with a warm structure, profile (**d**) shows the summer inverse surface anomaly with a cold surface and a warm subsurface structure, while panel (**e**) zooms into the 100 first meters of (**d**) to portray the SST inversion.

Due to the deep winter mixed layer, the Warm Core SST anomaly extends down to 250 m (Figure 9c). On this winter profile, the core of the anticyclone is always warmer that its surrounding down to 1000 m. An inversion of the eddy-SSTA is visible on the summer profile (Figure 9d). However, this Cold Core temperature anomaly does not extend below a few tens of meters (Figure 9e). Below the strong summer stratification, at −100 m for instance, the core temperature of the anticyclone is warmer than its surrounding waters

whose temperature is portrayed by the mean outside-eddy profile (black line in Figure 9d). The subsurface temperature anomaly reaches a positive value of $+1\,°C$ at 500 m, which is coherent with other observations of long-lived anticyclonic eddies in the region [24,45]. Hence, these unique in situ measurements indicate that the inverse eddy-SSTA remains confined to a few dozen meters below the ocean surface and does not correspond to the deep subsurface heat content of the anticyclone.

### 3.2. A Mechanism of SST Anomaly Inversion: Single Column Simulations

The analysis from the perspective of the regime change of individual eddies between Warm and Cold Core portrays that the winter mixed layer varies significantly inside long-lived mesoscale eddies, particularly in the core of anticyclones. However, is this MLD difference between the core of the eddy and its vicinity sufficient enough to explain the inverse eddy-SSTA that occurs during the spring re-stratification?

To answer this question and investigate other hypotheses, we use a simplified 1D model of the vertical advection–diffusion of heat in a stratified water column inside and outside mesoscale eddies. The seasonal forcing of the atmosphere is simulated with a sinusoidal surface temperature flux as $Q = Q_0 sin(2\pi t/365.25)$ ($Q_0 > 0$, positive for ocean heat gain), so that the simulation starts with a temperature flux increasing from zero (corresponding to spring). A value of $Q_0 = 150\,W/m^2$ is chosen as an accurate Mediterranean average of total surface heat flux seasonal cycle, following [46], with a negative (positive) maximum heat flux approximately in December (July). As salinity effects are neglected, the temperature flux is equivalent to the buoyancy flux.

An equal surface heat flux is applied for different temperature profiles corresponding to water columns inside a cyclone, anticyclone and a profile outside an eddy, respectively. The simulation starts on the end of the winter period when the MLD is at its deepest. The initial profiles are described analytically in Equation (3), whose parameters are chosen so that the simulated profiles represent average temperature profiles in the Mediterranean sea. These stand for a homogeneous MLD of $Z_{MLD} = 50\,m$ at $T_0 = 16\,°C$ for a cyclone core, 200 m at 18 °C for an anticyclone core and 100 m at 17 °C for the outside-eddy profile. Below the mixed layer, we introduce an exponential decrease (typical thickness $Z_T = 150\,m$) to a deep ocean value of $T_b = 13.5\,°C$. The $T(z)$ profiles are plotted on panels (b) and (d) of Figure 10, with a blue, red and black line for the anticyclone, cyclone and outside-eddy profiles, respectively.

$$
\begin{aligned}
T(z) &= T_0 &&; \text{if } z > Z_{MLD} \\
T(z) &= T_b + (T_0 - T_b)exp\left(\frac{z - Z_{MLD}}{Z_T}\right) &&; \text{if } z < Z_{MLD}
\end{aligned}
\tag{3}
$$

Vertical profiles are forced by the surface heat flux, and in a one-dimensional space, the temperature temporal evolution follows a simple diffusion equation:

$$
\frac{\partial T}{\partial t} = \frac{\partial}{\partial z}\left(A(z)\frac{\partial T}{\partial z}\right)
\tag{4}
$$

We assume a steady turbulent mixing coefficient $A(z)$ which depends only on depth. This vertical mixing profile is set by the Equation (5), starting from a surface value $A_0$ down to a deep ocean value $A_{back}$ with a Gaussian vertical shape, with $Z_T = 150\,m$. The static instability (i.e., $\partial_z T < 0$) is inhibited by simply boosting the mixing coefficient $A(z)$ to $1\,m^2/s$.

$$
A(z) = A_{back} + A_0 e^{-(z/Z_T)^2}
\tag{5}
$$

We first assume that the vertical mixing profile remains the same in the cyclone, the anticyclone and the outside-eddy. The surface value $A_0$ of $10^{-4}\,m^2s^{-1}$ is chosen as a typical value for turbulent mixing in the upper ocean, while in the deep ocean, the mixing is

reduced by two orders of magnitude down to $A_{back} = 1.0 \times 10^{-6}$ m$^2$s$^{-1}$, which is the water kinematic viscosity.

The uniform vertical mixing profile, common for both three water columns, is plotted in panel (a) of Figure 10, while the response of the three water columns (anticyclone, cyclone, outside-eddy) is plotted in panel (b). The simulation starts from a deep-MLD profile at the end of the winter mixing period (dashed line). During spring re-stratification, the positive surface is transferred downwards at the same rate for all water columns. As a consequence, the surface temperature difference between the three winter profiles is also maintained in summer (continuous line). This effect is also observed in panel (b) of Figure 11 where the seasonal evolution of the SST of the three water columns is plotted on a 18-month period. The anticyclonic (cyclonic) profile is constantly warmer (colder) than the outside-eddy profile. A two-month lag between the surface flux of Figure 11 panel (a) and the SST of panel (b) is explained through the delay needed for the water column to integrate the radiative forcing.

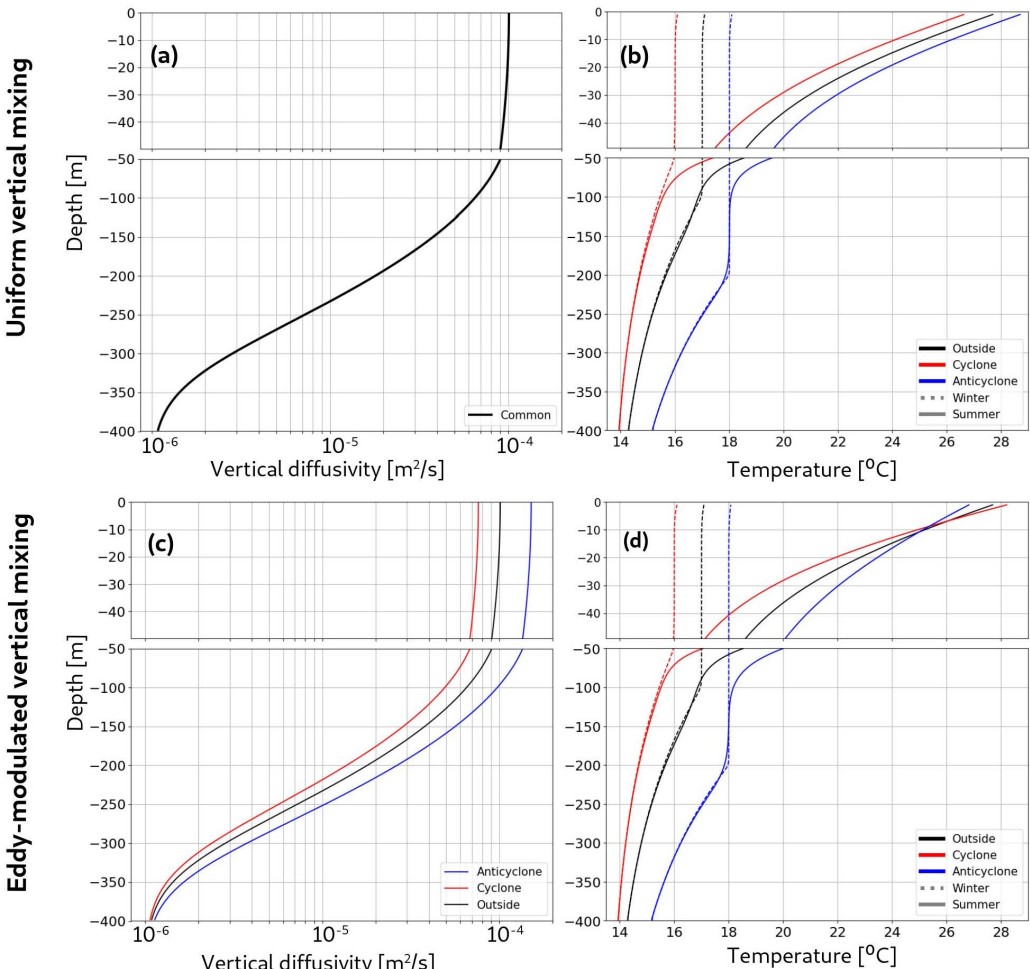

**Figure 10.** One-dimensional (1D) single column experiments: vertical structure. With a uniform vertical mixing: (**a**) vertical diffusivity $A(z)$ from Equation (5) and (**b**) temperature profiles in winter (dashed line) and summer (continuous line), corresponding time of the year being reported on Figure 11b . Initial winter profiles are set in Equation (3). With eddy-modulated vertical mixing: (**c**) vertical diffusivity and (**d**) temperature profiles.

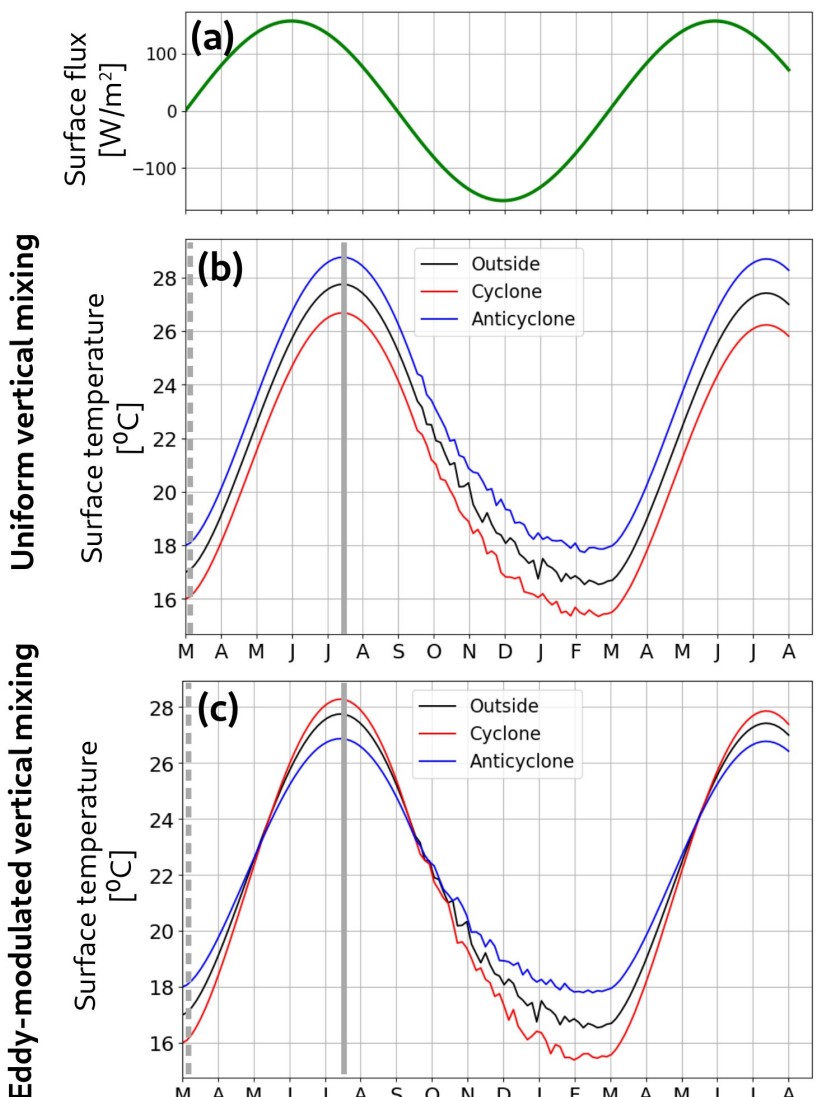

**Figure 11.** One-dimensional (1D) single column experiments: surface temperature. (**a**) Surface heat flux forcing the simulation, (**b**) Surface temperature evolution for anticyclone, cyclone and outside-eddy profiles with a uniform vertical mixing and (**c**) same as (**b**) but with an eddy-modulated vertical mixing, as shown in Figure 10c.

The initial differences of temperature profiles and winter MLD are not sufficient to reproduce observed eddy-SSTA inversion, suggesting that an additional physical process is missing. To explore such a mechanism, we assume that the vertical mixing coefficient is, on the one hand, enhanced in the core of anticyclonic eddies and, on the other hand, reduced in the core of cyclonic eddies. To test this hypothesis, we perform another set of heat vertical diffusion experiments with the same vertical profiles (Equation (3)) and diffusion process (Equations (4) and (5)) but with a varying surface vertical diffusivity value. $A_0$ stays outside-eddy at $1.0 \times 10^{-4} \, \mathrm{m^2 s^{-1}}$ but is divided by a factor of 2 to $5.0 \times 10^{-5} \, \mathrm{m^2 s^{-1}}$ in the cyclone profile and multiplied by 2 to $2.0 \times 10^{-4} \, \mathrm{m^2 s^{-1}}$ in the anticyclone one. These eddy-modulated vertical mixing profiles are plotted in panel (c) of Figure 10 with a blue/red/black colour representing the anticyclone/cyclone/outside-eddy profile.

Through the seasonal evolution results of the eddy-modulated vertical mixing model, as shown in panel (d) of Figure 10, it is observed that heat penetrates deeper in the vertical structure of the anticyclone, resulting in a less stratified profile in summer (blue continuous line). Vice versa, the heat reaches a shallower depth of the cyclone, resulting in a more stratified summer profile (red continuous line). The vertical diffusivity difference is strong

enough that the anticyclone (respectively cyclone) profile becomes cooler (warmer) than the outside-eddy profile, resulting in an isotherm crossing similar to what was observed in the Ierapetra anticyclone, which is seen at panel (e) of Figure 9.

The evolution of surface temperature given by the eddy modulated vertical mixing model, shown in panel (c) of Figure 11 for more than a year and a half, reproduces the same SST anomaly summer inversion in cyclones and anticyclones. The column representative of an AE (CE) core is indeed warmer (colder) in winter at the surface than a column representative of an outside-eddy profile stratification while turning colder (warmer) in summer, implying that an inverse SST anomaly has emerged.

These simplified 1D model simulations show that despite initial differences in vertical stratification or MLD, a differential mixing coefficient between the core and the periphery of oceanic eddies is needed to explain the inverse sea surface temperature anomalies which emerge during the spring re-stratification period.

## 4. Discussion

The emergence of inverse eddy SST anomalies during the summer season, in the global ocean as well as in regional seas, has been well documented by recent studies [30–33,35]. Some of them also link this inversion of the eddy surface anomaly with the spring re-stratification of the ocean surface. This study confirms that such seasonal correlation is also valid for the Mediterranean Sea (Figure 6). Nevertheless, we showcase here that the difference in the MLD alone is a necessary but not sufficient condition for the emergence of an inverse eddy SST anomaly. We consider the hypothesis that eddies modulate the diapycnal mixing in their interior, creating a vertical spacing (tightening) of isopycnals in anticyclones (cyclones). Our 1D single column modelling experiment (Figures 10 and 11) shows that a modulation of vertical mixing $A(z)$ inside eddies is needed to reproduce the inversion of the eddy-induced SST anomalies during summer. The origin of this vertical mixing modulation might stem from 3D dynamical processes that cannot be reproduced explicitly in the 1D vertical model.

Some hypotheses exist in the bibliography, and several papers studied the trapping of Near-Inertial Oscillations (NIO) in anticyclones either through a theoretical formulation [47,48] or through modelling experiments [49–51] and observations [52]. Enhanced turbulent mixing at depth was also reported when NIO were trapped inside anticyclones [53,54]. Nevertheless, we are not aware of a theoretical study that provides a direct link between the trapping of NIO and enhanced vertical mixing in the thermocline layer. The opposite trend for cyclones is still under discussion. However, due to the Coriolis effect $f_{eff} = f + \zeta$, which is higher for positive vorticity area ($\zeta > 0$), the spectrum of NIO is expected to be reduced in the core of cyclonic eddies [47,48]. This impact of NIO within the eddies is a very plausible explanation of the differential vertical mixing and the observed eddy-SSTA asymmetry between cyclone and anticyclones. Nevertheless, other mechanisms could also be responsible for inverse eddy SST anomalies such as nonlinear wind-induced Ekman pumping.

Motivated by the impact of eddies on biological productivity, several studies investigate the self-induced Ekman pumping in the core of mesoscale cyclones and anticyclones. Local currents induced by coherent eddies generate a curl to the surface stress from the relative motion between surface air and water, even if the wind stress is uniform. Hence, a uniform wind applied to an anticyclonic eddy can lead to a divergence and upwelling in its core [9,28,55]. A local upwelling could therefore induce a Cold Core anomaly for anticyclones. However, such a mechanism depends directly on the eddy intensity, and we did not find on the data of this study any correlation between the amplitude of the temperature anomaly in the core of the eddy and its intensity. Nevertheless, to investigate in more details the impacts of the wind-induced Ekman pumping on the emergence of inverse eddy SST anomalies, a full 3D numerical simulation will be performed in a future work.

## 5. Summary and Conclusions

The emergence of inverse eddy SST anomalies in the Mediterranean Sea is a seasonal phenomenon that affects all mesoscale eddies. Remote sensing and in situ observations were used to detect and quantify the eddy-induced SST anomaly over a 2-year period (2016–2018). We build an eddy core SST anomaly index to quantify the amount of Cold Core Anticyclones and Warm Core Cyclones all over the year and especially during the spring re-stratification period. We find that these inverse eddy anomalies could reach a peak of 70% in May and June. This seasonal cycle coincides with the seasonal variation of the MLD both through a statistical analysis, on a basin scale, and through an individual analysis for long-lived eddies. By tracking these eddies, we find that some of them alternate several times, from one season to another, between a Warm Core and a Cold Core SST anomaly. However, the inverse eddy anomalies are constrained to the upper layer of the ocean. For instance, co-localising ARGO profiles in Cold Core anticyclonic eddies reveals that their cold temperature anomaly inversion is limited to the first 50 m of the ocean, while a warm subsurface anomaly persists deeper.

We propose a simple dynamical mechanism, based on a differential mixing between the eddy core and its surroundings, that leads to reproducing Cold Core (Warm Core) anticyclones (cyclones) during the spring re-stratification period. To do so, we construct a simple vertical column model to compute the impact of the seasonal air–sea flux on the vertical stratification inside and outside eddies. We start off by a winter stratification setup with a deep mixed layer and investigate how the spring re-stratification of the ocean surface differs between the eddy core and its surrounding. By considering only the MLD difference, we were not able to reproduce the inverse eddy-SSTA that are observed during the spring re-stratification period in satellite data. It is only by taking into account a differential diapycnal eddy mixing—increased in anticyclones and diminished in cyclones—that we reproduce correctly the surface temperature inversion in the eddy core with respect to an outside-eddy profile. This simplified model suggests that vertical mixing modulation by mesoscale eddies might be the key mechanism that leads to the eddy-SSTA seasonal inversion in the ocean. Even if our study focuses on the Mediterranean Sea, the mechanism proposed here is, a priori, independent of the oceanic region.

Several theoretical studies on near inertial oscillations and corresponding in situ observations could explain the modulation of the vertical mixing induced by oceanic eddies and the cyclone/anticyclone asymmetry. However, full 3-dimensional modelling is necessary to further investigate these dynamical modes in combination with the wind-induced Ekman pumping inside the eddy core. Such high-resolution simulations are beyond the scope of this study and will be the perspectives of a future work.

Lastly, this study showcases that a detailed analysis of remote sensing observations of the complex eddy signature at the ocean surface could reveal its subsurface structure in the first tens of metres. This would provide valuable information on the vertical extension of the mixing layer or the bio-geochemical activity in the euphotic layer.

**Author Contributions:** Conceptualization, E.M., A.B. and A.S.; methodology, E.M, A.B. and A.S.; software, E.M. and A.B.; validation, A.S.; formal analysis, E.M., A.B. and A.S.; investigation, E.M., A.B. and A.S.; resources, A.S.; data curation, E.M. and A.B.; writing—original draft preparation, E.M. and A.B.; writing—review and editing, A.S.; visualization, E.M. and A.B.; supervision, A.S.; project administration, A.S.; funding acquisition, A.S. All authors have read and agreed to the published version of the manuscript.

**Funding:** This research received no external funding.

**Data Availability Statement:** This study has been conducted using E.U. Copernicus Marine Service Information; https://doi.org/10.48670/moi-00141, https://doi.org/10.48670/moi-00171. The DYNED-Atlas of eddy tracks in the Mediterranean Sea is publicly available: https://www1.lmd.polytechnique.fr/dyned/.

**Conflicts of Interest:** The authors declare no conflict of interest.

## Appendix A. Eddy Timelines

We provide three additional eddy timelines of long-lived eddies: an Algeria anticyclone (Figure A1), a Cyprus (Eratosthenes) anticyclone (Figure A2) and a Liguria cyclone (Figure A3). The reader is referred to Figure 8 (timeline of the Ierapetra long-lived eddy) in the main corpus of the text for a detailed description of the timelines as individual eddy analysis of the sea surface temperature anomaly evolution.

We note several particularities compared with the Ierapetra anticyclone: The Algeria anticyclone in the western Mediterranean shows the same swift between Cold Core anomaly (summer) to Warm Core anomaly (winter) while having a smaller local MLD than the Ierapetra eddy. The Cyprus anticyclone in the eastern Mediterranean shows a persistent inverse Cold Core anomaly ranging from May to December, while the regular Warm Core anomaly appears only for a few winter months. Finally, the Liguria cyclone shows that the shift between regular and inverse anomalies can also emerge for long-lived cyclones, corresponding also with the MLD seasonal cycle.

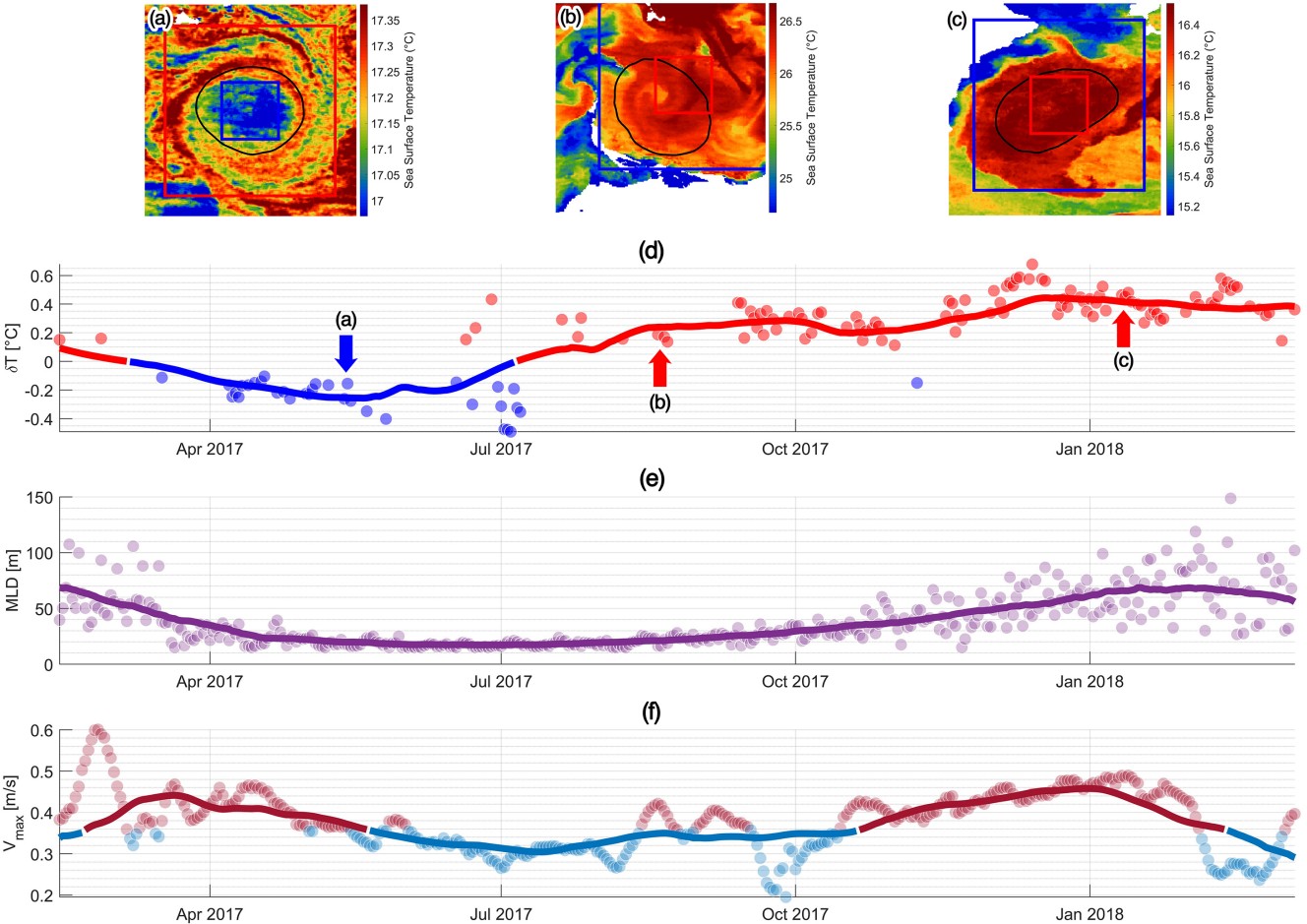

**Figure A1.** Timeline of the Algeria anticyclone with DYNED ID #11411. Panels (**a**–**c**) show four characteristic SST patches corresponding with the $\delta T$ evolution in panel (**d**). Panel (**e**) shows the evolution of the MLD outside the eddy. Panel (**f**) shows the evolution of the maximal velocity. Lines represent the Monthly Mean Average of each variable. For more information, the reader is referred to Figure 8.

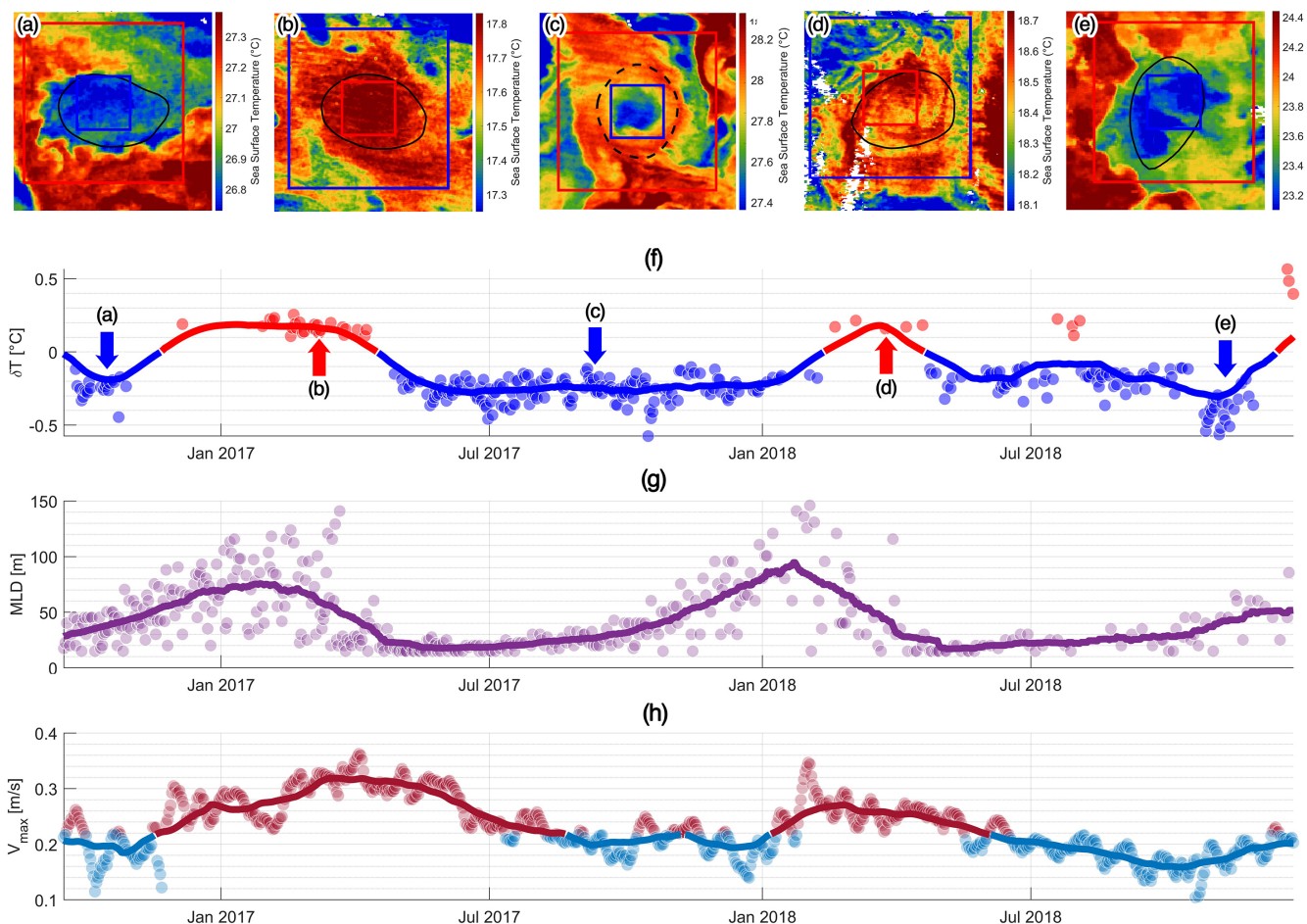

**Figure A2.** Timeline of the Cyprus (Eratosthenes) anticyclone with DYNED ID #10754. Panels (**a**–**e**) show four characteristic SST patches corresponding with the $\delta T$ evolution in panel (**f**). Panel (**g**) shows the evolution of the MLD outside the eddy. Panel (**h**) shows the evolution of the maximal velocity. Lines represent the Monthly Mean Average of each variable. For more information, the reader is referred to Figure 8.

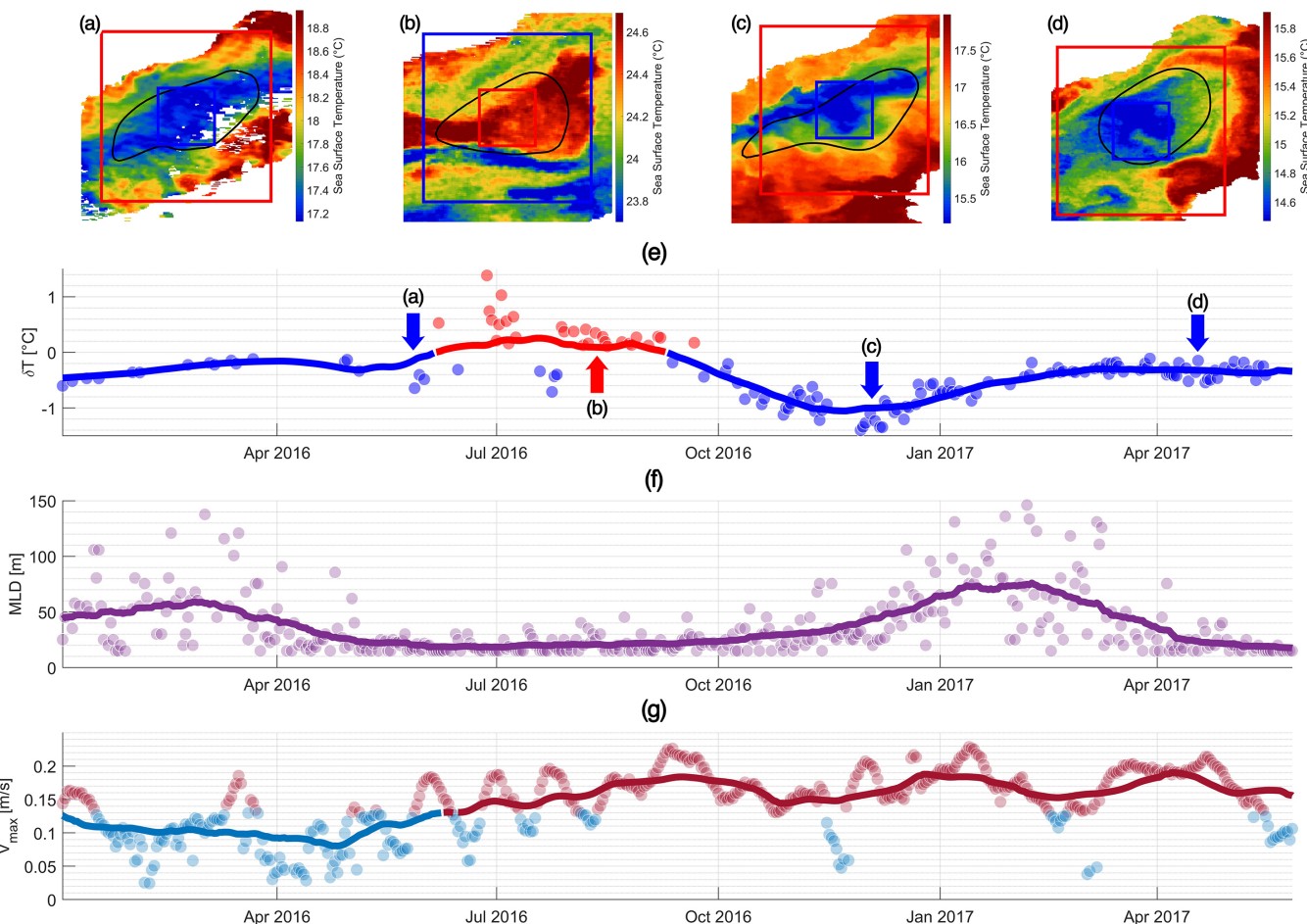

**Figure A3.** Timeline of the Liguria cyclone with DYNED ID #9784. Panels (**a**–**d**) show four characteristic SST patches corresponding with the $\delta T$ evolution in panel (**e**). Panel (**f**) shows the evolution of the MLD outside the eddy. Panel (**g**) shows the evolution of the maximal velocity. Lines represent the Monthly Mean Average of each variable. For more information, the reader is referred to Figure 8.

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
