# Peer review of "Why Do Inverse Eddy Surface Temperature Anomalies Emerge? The Case of the Mediterranean Sea"

_remotesensing, doi:10.3390/rs14153807_

Round 1

Reviewer 1 Report

The following is the review for:

Why do inverse eddy surface temperature anomalies evolve: The case of the Mediterrean Sea.

Summary: The paper focuses on the application of remote sensing to try and determine why eddies exist that apparently have temperature anomalies contradictory to those anaiticpated with cyclonic and anticyclonic gyres. For example cyclonic eddies should be associated with colder temperatures in the core of the eddy while anticyclonic eddies should be associaited with warmer temperatures in the core. The overall conclusion of the paper is that the anomalies are seasonal and are apparently associated with processes associated with mixing and stratification.

Strengths:

The primary strength of the paper is how they incorporate, satellite, in-situ (ARGO) and models to explain a physical phenomena found in the ocean. Based on this alone I recommend publication with perhaps a minor revision.  Overall the paper is well written and organized.

Comments:

The authors (hopefully won’t be too difficult) need to implement the Remote Sensing template. For example their needs to be an explicit “Materials and Methods” section. Some other comments that the authors could address in a revision:

1)    Can the authors comment on the possible impact of cloud contamination on results. For example the number of eddies detected?  Assume that cloud contamination could prevent eddies from being diagnosed?  This illustrated in Figure 3. One question would be if clouds prevent the exact location replicated in the SST fields could this impact the derivation of the anomaly? The authors explain a filtering technique that was applied. Could they comment how this might impact the results, especially the seasonal analysis, etc.

2)    Can the authors give more detail on the SST data used? What were the input sensors used in the composite imagery?

Author Response

Thank you for your rapid and comprehensive review of our manuscript. 

Your proposals have been incorporated in the reviewed manuscript.

Please find attached a pdf with responses to several points.

Please excuse a short delay due to attendance to the IGARSS conference.

Reviewer 2 Report

Review of “Why do inverse eddy surface temperature anomalies emerge? The case of the Mediterranean Sea” by Moschos et al.

General comments:

It is showed in this study that the existence of inverse eddy SST anomalies is a seasonal phenomenon that affects the life cycle of mesoscale eddies in the Mediterranean Sea. Both remote-sensing observations and in-situ data were used in this analysis. A 1-D vertical mixing model is used to interpret this inverse eddy SST phenomenon. The work is very interesting, and the results are convincing. The figures are well-designed and beautifully plotted. I enjoyed reading the manuscript. My main criticism is about the writing – there are many minor technical errors, which should be easy to fix if the authors invest enough time.  I have already listed several as below, but it is authors’ responsibility to fully sort out this kind of issues. So, I would like to recommend the manuscript be accepted after some minor revision.

Specific comments:

Line 7, “surface temperature index” should be more specific, “SST anomaly index”.

Line 8, “amount” should be changed to “strength of SST anomaly inversion”.

Line 52, “MLD” should be defined as it first appears in the text.

Lines 86-87, “MLD” should be used instead of “mixed layer depth”.

Line 106, “SST” has already been defined previously.

Line 110, “MLD” has already been defined previously.

Line 167, Warm-Core (WC) or Cold-Core (CC) should not be abbreviated.  Because “CC” can be easily regarded as “counter-clockwise” for eddies, and cause confusion to readers.  Too many acronyms make a paper unnecessarily hard to read.  Readers may have to go back multiple times to check what an acronym stands for.

Lines 178-179 needs some work, “in order to For the calculation of the”?

Figure 3 caption, “(c) and (d) are filtered” should be changed to (c) and (d) are discarded”.

Line 205-206, it is not clear how altimetry SSH was used in this analysis. Maybe I missed it.

Line 211, CCP should be clearly defined.

Lines 234-235, is this patch referred to the small patch (Rmax) or the larger patch (5Rmax)?

Line 239, “MLD” has already been defined previously.

Line 244, “On winter” should be changed to “In winter”.

Line 246, “On summer” should be changed to “In summer’. Also, “however” should be moved to the beginning of the sentence.

Line 247, it would be good to added a sentence to support this feature: “This is consistent with the findings in the Gulf of Mexico that SST tends to be more spatially uniform in summer than in winter (Liu et al., 2011, doi:10.1029/2011GM001127)”.

another reference also be suggested: doi:10.1029/2003GL017673

Line 250, “thus” should be removed.

Line 253, “at” should be changed to “in”.

Line 254, 263, and many other places, it should be consistent throughout the text for the use of “cold-core”, “Cold-Core”, “code core”, “Cold Core”.  Same for “warm-core”, and maybe other words.

Line 301, “MLD” has already been defined previously.

Line 303, “MMA” should not be abbreviated. It does not save much space, rather it causes unnecessary inconvenience in reading.

Line 453, Line 250, “thus” should be removed or moved the beginning of the sentence.

Line 466, “the effective Coriolis parameter” should be changed to “the Coriolis effect”.

Author Response

(The authors gave the same response as above.)

Reviewer 3 Report

Why do inverse eddy surface temperature anomalies emerge? The case of the Mediterranean Sea by Moschos et al.

This study tried to explain the inverse anomaly mechanism. Even though it is not very persuasive, it is interesting. I suggest publication after some modification.

Main points:

1.       It is suggested to give brief descriptions of the forming mechanism of the regular eddies, so the people who are not the expert like me in this field can grasp the main picture easily.

2.     The DYNED-Atlas data are used here. Have you compared with the Aviso products?

Minor points:

1.     Lines 46-53: It looks strange to me that a sentence starts with a reference number, it may be better to change the sentence structure.

2.     V, Lp, A in equations 1 and 2 are not explained.

3.     Line 321  on winter months   ? in winter months

4.     Fig. 9, caption: 26 February 2017 and 26 June 2017 are described in the text, but only 26/07/2017 is noted in the caption. Please check if there is an error.

Author Response

(The authors gave the same response as above.)
